# Developing a dynamic model of anomalous experiences and function in young people with or without psychosis: a cross-sectional and longitudinal study protocol

Abigail C Wright,[1,2] David Fowler,[1,2] Kathryn E Greenwood[1,2]

¹School of Psychology, University of Sussex, Brighton, UK
²Research & Development, Sussex Partnership NHS Foundation Trust, Worthing, UK

**Correspondence to**
Miss Abigail C Wright;
a.c.wright@sussex.ac.uk

## ABSTRACT

**Introduction** Anomalous experiences are common within the general population, but the frequency and intensity is increased in young people with psychosis. Studies have demonstrated that perceptual biases towards noticing these phenomena plays a role, but the way one thinks about one's experience (metacognition) may also be relevant. While poor metacognitive function has been theoretically associated with anomalous experiences, this relationship is currently unclear. However, metacognition may work along a continuum with various metacognitive levels, many of which have been demonstrated as impaired in psychosis. These metacognitive components may interact via processes that maintain poor metacognition across levels, and that potentially impact both what people do in their everyday lives (functional outcome) and how people feel about their everyday lives (subjective recovery outcome) in young people with psychosis compared with healthy control participants.

**Methods and analysis** This study will investigate the association and contribution of metacognition to anomalous experiences and outcome measures cross-sectionally and longitudinally in a 36-month follow-up. First, young people with psychosis will be compared with healthy control participants on selected measures of anomalous experience, metacognition, and function, using analysis of covariance to identify group differences. Next, the relationship between metacognitive components and processes will be explored, including processes connecting the different components, using regression analyses. Finally, mediation analyses will be used to assess the predictive value of metacognitive measures on outcome measures, both cross-sectionally and longitudinally at 36 months, while controlling for symptoms and cognition.

**Ethics and dissemination** Ethical and Health Research Authority approval has been obtained through Camberwell St. Giles Research Ethics Committee (reference number: 17/LO/0055). This research project will be reported within a PhD thesis and submitted for journal publication. Once key predictive components of poor outcome in psychosis are identified, this study will develop a series of dynamic models to understand influences on outcome for young people with psychosis.

## Strengths and limitations of this study

► This is one of the first studies to assess metacognitive processes as predictors of functional outcome in young people with psychosis, and one of the first to assess metacognition as a longitudinal predictor of functional outcome in young people with psychosis sample.

► This study has used up-to-date paradigm within the field of metacognition to avoid biases.

► Due to length of follow-up period, it may be difficult to engage follow-up participants within this study.

► If recontacting is difficult, the study may only have power to detect effects of the separate paths of the longitudinal analyses, not the full model.

## INTRODUCTION

Anomalous experiences refer to a rich number of various psychic phenomena. These experiences can be divided into three main categories: anomalous self-experiences, the sense that you are not 'real' (distortions in experience of self and being); anomalous perceptual experiences, hearing sounds which cannot be accounted for by the environment (distortions of sensory events in various domains; auditory, visual; touch; taste); and anomalous delusional beliefs, experiencing unusual thoughts or beliefs. These experiences may be common within the general population[1 2] but the frequency and intensity of these anomalous experiences is increased in those with psychosis or those with emerging severe mental health difficulties[3 4] and may predict psychotic symptoms at a later stage.[5 6] Research has suggested that anomalous self-experiences are suggested to precede and generate 'surface-level' anomalous perceptual experiences (hallucinations)[7] and anomalous delusional beliefs may be developed from anomalous experiences,[8–10]

suggesting a hierarchical framework between the anomalous delusional beliefs, experiences and self-experiences.

Many theories have been proposed to understand anomalous experiences, including source-monitoring deficits[11] and aberrant salience hypothesis.[12] Signal detection theory (SDT) has been the foundation to this research and studies using SDT have demonstrated that anomalous perceptual experiences are associated with perceptual signal detection biases[13 14]; bias towards stating that a stimulus was present when it was in fact absent. Such signal detection biases have been consistently shown within psychosis/psychosis-proneness literature[15–17] and suggest that top-down processes on (false) perception can lead to hallucinations or delusions.[8 18 19]

Recent evidence suggests these signal detection biases are associated with metacognition in healthy students,[20] and metacognition may therefore play a role in anomalous experiences. Metacognition is defined as 'thinking about thinking',[21 22] an abstract view of the object level.[23] Literature has demonstrated that hallucinatory experiences and delusional beliefs/ideation have been associated with metacognition (overconfidence, specifically) in both clinical and non-clinical groups.[24–26] In particular, those with psychosis demonstrate more incorrect self-monitoring responses with higher confidence,[27 28] also present in those with a history of hallucinations,[29] and those at high-risk groups.[30]

This research has not been consistent (see Gawęda *et al*[28]) as some studies did not control for objective performance, crucial for metacognitive efficiency scores.[31 32] A recent controlled study demonstrated that individuals with first-episode psychosis (FEP) and those at risk were more likely to misattribute an imagined action for a performed action compared with healthy controls,[30] but found no difference in misattribution of verbal or non-verbal actions, which suggests the deficit in metacognition may be across several modalities. However, metacognitive efficiency/sensitivity (measured using meta-*d'*) is known to be modality-specific[33 34] and anomalous perceptual-experiences (eg, hallucinations) can vary in modalities.[35] It has also been acknowledged that auditory anomalous experiences are most common in psychosis,[36 37] all of which may suggest a modality-specific association with auditory or visual anomalous experiences and perception/metacognition. This present study will

assess the modality-specific association between perceptual bias (signal detection bias) and metacognitions with anomalous perceptual experiences in visual and auditory modalities, while controlling for objective performance (see figure 1).

Limited research has assessed the association between anomalous self-experiences and metacognition, but it may be suggested metacognitive efficiency may also be associated with anomalous self-experiences, previously alluded to by Dokic and Martin.[38] This study will empirically test the association between anomalous self-experiences and perceptual biases and metacognition. Anomalous self-experiences and anomalous delusional beliefs have not been considered to be modality-specific; therefore, these measures are hypothesised to be related to both visual and auditory perceptual signal detection biases and metacognitive ability.

Metacognition has been considered fractionated and can appear in many different forms, associated within a dynamic model.[23 39] Three levels of metacognition have been proposed: (i) metacognitive efficiency: 'knowing that you know',[40] and this level in particular may involve unconscious knowledge to generate a 'feeling of knowing'[41] and has been shown to be modality-specific,[33] which can be assessed by within-task confidence ratings; (ii) metacognitive experience: appraisal of one's experience or performance after an activity; and (iii) metacognitive knowledge/ability: capacity to think about one's or others' experience or abilities, on which Lysaker and colleagues have grounded their work (see Lysaker *et al*). These metacognitive levels may influence each other via metacognitive processes. Nelson and Narens[23] highlighted two processes: (i) metacognitive controlling processes (ie, such that knowledge is used to control, guide and correct ongoing action)[42] and (ii) metacognitive monitoring processes (ie, monitoring of ongoing experience in order to recognise anomalies and update higher level beliefs),[43] which are important for accurate metacognitive functioning.

As metacognition works in a hierarchical fashion, it may be expected the poor metacognitive efficiency in psychosis, demonstrated above, can impact the next component on the continuum: metacognitive experience, via metacognitive processes. However, Gilleen *et al*[44] demonstrated that metacognitive experience is on

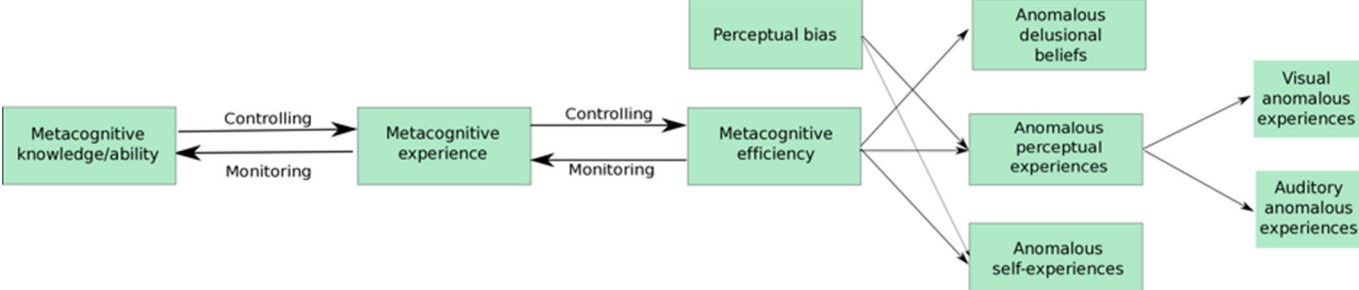

**Figure 1** Proposed model including associations between perceptual signal detection bias and metacognitive efficiency with various anomalous events: anomalous experiences and beliefs, and the associations between metacognitive aspects.

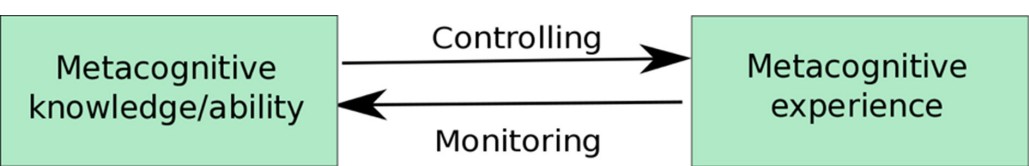

**Figure 2** Part of figure 1 model. Proposed associations between metacognitive efficiency and metacognitive experience with monitoring and controlling processes.

average intact within a group of individuals with schizophrenia, while metacognitive knowledge/ability is deficit. The dissociation between the different levels of metacognition may be the result of impaired metacognitive processes (self-reflectiveness and set-shifting),[45] akin to various other studies.[46–48] Due to limited studies, it is currently unclear which metacognitive processes may be driving these metacognitive difficulties. This study aims to further understand the difficulties in metacognitive processes in young people with psychosis and matched healthy controls within two models (figures 2 and 3).

From this, we will assess which metacognitive components may impact on functional outcome: a measurable aspect of an individual's specific activities of daily living. Research suggests functional outcome is predicted by neurocognition,[49 50] functional capacity (measured using real-life performance skills task),[51–53] negative symptoms,[54] which has been demonstrated to show a synergistic interaction with cognition to impact functioning,[55] and, importantly here, metacognition. Metacognitive ability, measured using the Metacognition Assessment Interview (MAI) or MAS,[22 56] appears to play a crucial role on functional outcome, independent of cognition and symptoms.[57–59] In a recent study, Davies *et al*[60] demonstrated that metacognition partially mediates the relationship between cognition and functional capacity, and fully mediates the relationship between functional capacity and functional outcome. Limited research has assessed whether metacognitive processes are also relevant in maintaining poor functional outcome over time, independent of cognition. With this in mind, a metacognitive factor will be derived from all metacognitive variables above to explore the impact on functional outcome and capacity (see figure 4) and a longitudinal model will also explore the relationships over time.

Functional outcome has also been associated with subjective recovery outcome (self-rated outcome reflecting sense of well-being and quality of life).[61 62] Metacognitive capabilities were related to components of recovery beyond the effects of psychiatric symptoms, including aspects of quality of life.[63–65] There is a complex

relationship between metacognition, functional outcome and subjective recovery outcome. This study will assess these relationships to enable in-depth understanding of functional recovery.

This study aims to develop and test a series of dynamic models to understand (i) the nature of metacognitive deficits compared with healthy controls, (ii) the relationship between metacognitive components (iii) and the influences of metacognition on anomalous experiences and objective/subjective functional outcome for young people with psychosis. If these proposed models can be demonstrated empirically, this can help to understand and remediate poor outcome within psychosis.

## Hypotheses

**Hypothesis 1**: Anomalous experiences will be associated with increased signal detection biases and poor metacognitive efficiency, and this relationship will be domain-specific.

**Hypothesis 2**: Metacognitive control processes will significantly predict metacognitive knowledge/ability and metacognitive monitoring processes will significantly predict metacognitive experience. Metacognitive control processes will significantly predict metacognitive experience and metacognitive monitoring processes will significantly predict metacognitive efficiency.

**Hypothesis 3**: Metacognitive variables (metacognitive knowledge/ability, metacognitive processes and metacognitive experience) will significantly predict outcome measures (functional capacity, functional outcome and subjective recovery outcome) in young people with and without psychosis, even after controlling for anomalous experiences, symptoms and IQ.

**Hypothesis 4**: Metacognitive variables (metacognitive knowledge/ability, metacognitive knowledge processes and metacognitive experience) will significantly predict outcome measures (functional capacity and functional outcome) at 36-month longitudinal follow-up of participants in young people with psychosis, even after controlling for symptoms and IQ.

**Figure 3** Part of figure 1 model. Proposed associations between metacognitive experience and metacognitive knowledge with monitoring and controlling processes.

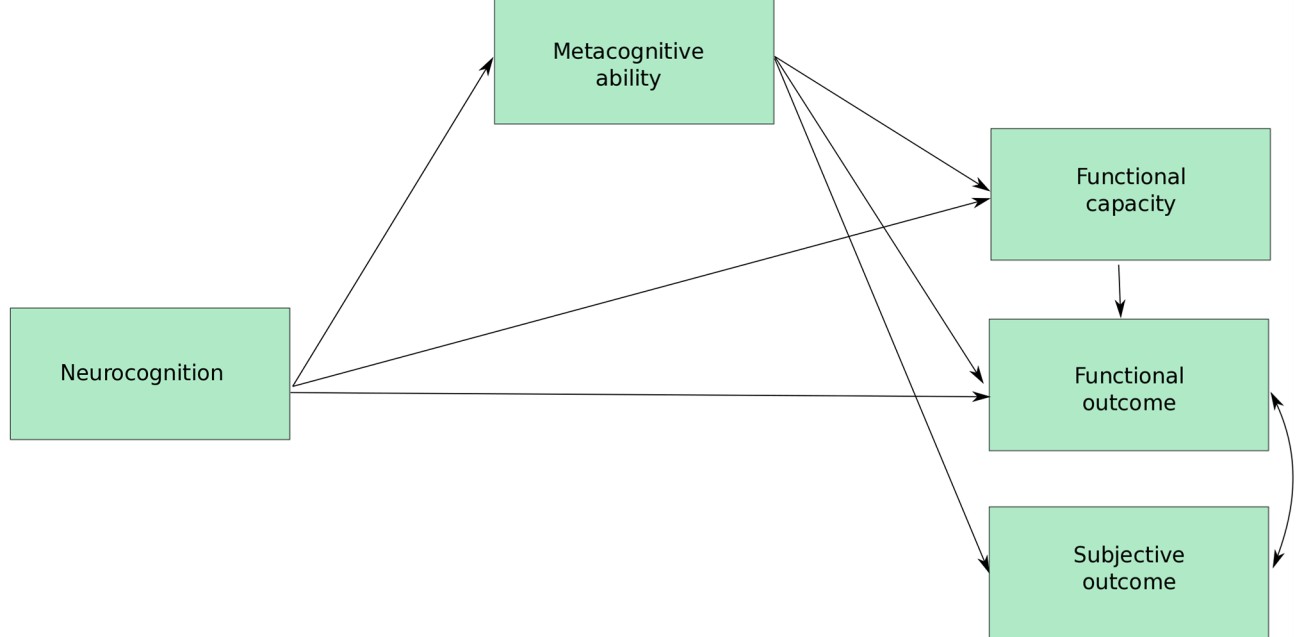

**Figure 4** Proposed model for cross-sectional and longitudinal analyses including indirect effects from symptoms and neurocognition to metacognition and functional capacity to functional and subjective outcome.

## METHODS
### Design
This is a cross-sectional and longitudinal study. This cross-sectional aspect will explore (i) deficits in metacognition between young people with psychosis and healthy controls; (ii) interrelations between different metacognitive components and processes across the whole sample and (iii) the contribution of specific novel metacognitive variables to anomalous experiences and outcomes and (iv) a longitudinal aspect will identify whether metacognition predicts experiences and outcomes at 36 month follow-up period (psychosis sample only).

### Participants
Seventy-three young people with psychosis will be recruited. This sample will be made up of a convenience sample from first episode services and the remaining individuals will be those re-recruited from a previous FEP sample (previous n=80), to take part in the main cross-sectional study and longitudinal follow-up aspect. Participants from the previous FEP study have baseline data on metacognition, functional capacity, functional outcome, symptoms and cognition.[60] These data will form the baseline data for the longitudinal analysis. All participants with psychosis will be 18–40 years of age, able to read and communicate in English and receive treatment for psychosis with a UK Early Intervention in Psychosis service at first assessment. Participants with organic causes for psychosis or those with a diagnosis of substance use disorder will be excluded.

Seventy-three healthy control participants will be recruited as a comparison group. These participants will be 18–40 years of age, able to read and communicate in English and matched on age, gender and education with the psychosis sample. Participants will be recruited through advertisement within the local community, for example, in libraries, cafes and on social media. Participants with current mental health problems or history of psychosis will be excluded following screening questions: (i) Are you currently experiencing any mental health difficulties? (ii) Are you on any psychotropic medication/ substances? (iii) Have you been in contact with psychological or psychiatric services for psychological problems? (iv) Has anyone in your immediate family experienced an episode of psychosis? For example, parents, siblings. If healthy control answered yes to any of the questions, these participants were deemed ineligible to take part in the study.

Any participant with hearing or sight problems which cannot be will be excluded. Data collection will be undertaken within a National Health Service (NHS) building or community setting between 10th March 2017 and 4 May 2018. The end date of the study is 18th September 2018.

### Patient and public involvement
Patient and public involvement is primarily via the Psychosis Interest Group run by the Service User and Carer Involvement Coordinator at the Sussex Partnership NHS Foundation Trust Research and Development department, and service users within the Psychosis Theme Group (PTG). The first author met with the PTG to consult on the development of this project, including the design, method and procedure of the project. The lived experience group have viewed all the measures, including the two main computer tasks (visual and auditory tasks), and provided extensive feedback which has been incorporated into this project. Study participants who consent to receiving the study results will receive these by post/email. The first author and

the PTG will continue to meet to consult on recruitment procedures, and on the plans for dissemination to service user groups following data analysis.

## Cross-sectional measures
### Anomalous experience measures
*Anomalous perceptual experiences: Multimodal Unusual Sensory Experiences Questionnaire.*[35] This measures anomalous perceptual/sensory experiences with six subscales. Both the full scale and subscales have been demonstrated as possessing good reliability (auditory r=0.72, visual r=0.72), good validity between clinical and non-clinical group (Cohen's d=0.96) and significance with all other anomalous experience scale, and internal consistency (auditory α=0.82, visual α=0.88). The auditory and visual subscales will be used for this study which each provide a combined score for presence and frequency of anomalous experiences.

*Anomalous self-experiences*: *Cambridge depersonalisation scale (trait and state versions).*[66] The trait version includes 29 items measuring frequency (never to all the time) and duration (few seconds to more than a week) of anomalous self-experiences over the last six months. It demonstrates high internal consistency (>0.6), good validity with other scales (r=0.8) and good reliability (α=0.89),[66] and is useful for assessing depersonalisation in a schizophrenia group.[67] The state version includes 22 items measuring anomalous self-experiences within-the-moment on a scale of intensity from 0 to 100. Subscales of the Cambridge Depersonalisation Scale (CDS) are 'alienation from surroundings', 'anomalous subjective recall', 'emotional numbing' and 'anomalous body experience'.

*Anomalous delusional beliefs*: *Schizotypal Symptom Inventory.*[68] This measure assesses residual psychotic symptoms, providing a total score with separate subscales for paranoia, anomalous experience and social anxiety.[69] This present study will use the paranoia subscale as a measure of anomalous delusional beliefs and the anomalous experience subscale as a measure of anomalous perceptual experiences, to confirm Multi-Modality Unusual Sensory Experiences Questionnaire data assessing anomalous perceptual experiences. This scale demonstrates high internal consistency (non-clinical sample α=0.87 and clinical sample α=0.92), good validity with other scales (Positive and Negative Syndrome Scale (PANSS) with total r=0.59 and paranoia r=0.6) and good test–retest reliability (0.85 for total and 0.6–0.84 for subscales).

## Metacognition
### Metacognitive efficiency
This encompasses separate computer-based visual and auditory detection task. The critical task in both paradigms is to make two forced-choice binary judgments of whether a visual or auditory stimulus (dot or tone) was present or absent (first judgment) within a noisy picture or presentation of white noise, and whether this was associated with high confidence or low confidence in the first judgment (second judgment).

The first judgment can be used to calculate a score for signal detection perceptual sensitivity and perceptual signal detection bias. *Perceptual sensitivity* ($d'$): the ability to correct report whether the stimulus (dot/tone) was either present or absent. SDT posits that detection-making involves depicting whether certain waveforms called signals may or may not be embedded within background 'noise', using internal responses.[70] *Perceptual bias* ($B$) is the tendency to report one decision over the other, for example, stating the stimuli was present when it was in fact absent. The internal responses from perceptual sensitivity score are then compared with a decision criterion (c) so all evidence above criteria elicit a response of 'present' compared with below the criteria elicits the response of 'absent'.[71] The perceptual sensitivity score and the decision criterion can be fitted to an empirical receiver operating characteristic curve, enabling us to assess perceptual biases in responses, for example, whether someone was more or less likely to report stimuli as being present based on a lower/higher decision criterion.

The second judgment can be used to calculate a score for metacognitive sensitivity and metacognitive efficiency. *Metacognitive sensitivity* (meta-$d'$) is the ability to discriminate between correct and incorrect judgements. Alike to the SDT above, metacognitive sensitivity is based on assessment of sensitivity (how well confidence ratings discriminate correct from incorrect trials) and response bias (overall propensity for reporting high confidence). However, this score from the second judgment must take into account first judgement performance.[72] Meta-$d'$ indicates the $d'$ that would have been predicted to result in the confidence rating assuming the SDT. Optimal metacognitive sensitivity is when perceptual sensitivity score is matched. Meta-$d'$ greater or less than $d'$ indicates metacognition is better or worse than $d'$.[34] *Metacognitive efficiency* is one's ability to discriminate between one's correct or incorrect perceptual decision, while taking into account objective performance.[32] This is calculated as meta-$d'$/$d'$. Metacognitive efficiency (meta-$d'$/$d'$) was chosen as a more robust form of perceptual metacognition, over metacognitive sensitivity (meta-$d'$) which was previously used by researchers.[73]

Performance on the first judgemnt will be held constant throughout the task and across participants (using a 1-up-2-down staircase procedure) to ensure that metacognitive sensitivity/efficiency is measured independent of task performance and produces valid scores.

Studies including signal detection tasks typically involve a large number of trials (~400), to avoid statistical bias and large variance in scores.[74] However, following a pilot study, we reduced the trials to 200, while maintaining reliable data. To ensure the feasibility of conducting this study within a clinical population, who may present difficulty with attention and concentration,[75] the two computer tasks have been developed from a pilot study within a healthy student population.[76]

 

Alongside this, participants will be asked to prospectively and retrospectively rate their performance on the detection tasks. These ratings will be used to assess metacognitive knowledge and metacognitive experience.

*Metacognitive knowledge*: The prospective rating ('How well do you think you will perform overall? For instance, if you think you will get it right every time, select 100 (all correct). If you think you will be correct none of the time, select 0 (none correct). You can select any value in between 0 and 100 to indicate what percentage you think you will correctly identify') will assess task-related metacognitive knowledge.

*Metacognitive experience*: The retrospective rating ('How well do you think you performed overall on the task? For instance, if you think you were right every time, select 100 (all correct). If you think you were correct none of the time, select 0 (none correct). You can select any value in between 0 and 100 to indicate what percentage you think you correctly identified') will assess task-related metacognitive experience.

### Metacognitive ability

This will be assessed using the Metacognitive Assessment Interview[56] to measure metacognitive ability/knowledge. This measure assesses the ability to understand 'the self' and 'the other'; termed as one multidimensional construct as 'metacognition'. This measure has demonstrated good inter-rater reliability and internal consistency ($\alpha$=0.90 for total metacognition), factorial validity and reliability (r=0.62–0.90).[56]

### Metacognitive processes (monitoring and control)

Monitoring processes will be assessed using self-reflectivity subscale of Cognitive insight scale.[77] This measure possesses good internal consistency ($\alpha$=0.68), convergent validity (with SUMD-A delusions, r=−0.67)[77] and test–retest reliability (r=0.90).[78] Studies have demonstrated this measure is appropriate to assess metacognitive monitoring.[45 79] Control processes will be assessed via set-shifting using trail-making task[80] part B-A. This measure possesses good internal consistency (TMT-A $\alpha$=0.39, TMT-B $\alpha$=0.71), convergent validity (with Task Switching Paradigm r=0.32)[81] and good reliability of other forms of TMT (r=0.78).[82] This measure is appropriate to assess metacognitive control processes.[45]

### Functional outcome

*Functional outcome*: Time Use Survey (adapted from Short[83]) provides a retrospectively rated objective measure for hours spent engaging in structured activity per week.[84] This measure has been used within an FEP sample to assess functioning.[85 86] This measure has good reliability (inter-rater reliability at 0.99),[85] coder reliability (89% accuracy),[87] good validity as differences in time use have been demonstrated between different stages of psychosis, representing social recovery in psychosis,[85] validity as TUS is comparable to studies using functioning measures.[69]

*Functional capacity*: The UCSD Performance-Based Skills Assessment[53] provides a total score for real-life performance skills based on simulated tasks. It was adapted to be applicable for UK participants. This measure demonstrates high internal consistency ($\alpha$=0.88), good validity with other scales (Direct Assessment of Functional Status (DAFS) r=0.86) and good test–retest reliability (r=0.91).[88]

*Subjective recovery outcome*: The Questionnaire of Process of Recovery[89] provides a score for an individual's subjective functioning (psychosis participants only). This scale has two subscales: intrapersonal items related to hope, empowerment, confidence and interpersonal related to connectedness with others, others help/care, reliance. This possesses good internal consistency (intrapersonal subscale, $\alpha$=0.94, interpersonal subscale, $\alpha$=0.77), construct validity (General Health Questionnaire (GHQ) total score: intra, r=−0.83 inter, r=0.52) and reliability (intra, r=0.87, inter, r=0.77).[89]

### Covariates

*Symptom severity*: This assesses symptoms of psychosis using PANSS[90] (clinical participants only), which is the mostly widely used standardised instrument for assessing symptom severity in schizophrenia.[91] This measure has demonstrated good reliability and validity, and appropriate inverse correlations between positive and negative subscales.[92] This measure has good internal consistency (agreement for PANSS items r=0.69–0.94), construct validity (between PANSS and Andreasen rating system, r=0.77) and reliability (inter-rater correlations for PANSS scales ranged from 0.83 to 0.87).[92]

*Cognitive ability: This includes verbal IQ*: Vocabulary task[93] is a measure of an individual's verbal knowledge and fund of information. This measure as good internal consistency (correlation with other cognitive measures range r=0.54–0.79), construct validity (with WASI-III, r=0.88) and reliability (0.90–0.89) and test–retest (0.88)[93]; and performance IQ: Matrix reasoning task[93] is a measure of individual's ability to mentally manipulate abstract symbols and perceive the relationship among them. This measure as good internal consistency (correlation with other cognitive measures range r=0.59–0.63), construct validity (with WASI-III, r=0.66) and reliability (0.88–0.96) and test–retest (0.76).[93]

### Longitudinal measures

For participants who have baseline data, this comprises of the following measures outlined above: metacognitive knowledge/ability and metacognitive monitoring processes; functional outcome and functional capacity; and all covariates.

## ANALYSIS
### Sample size

Two sample size estimates have been combined to ensure the analyses have sufficient power to detect effects.

First, sample size to detect differences between individuals with psychosis and healthy control participants on metacognitive efficiency. Using G power with 0.8 power, 0.57 effect size based on a previous metacognitive efficiency task in psychosis[94] and 0.05 alpha, the proposed total sample size is 28 (14 per group). Second, a regression analysis to assess the predictive value of metacognitive knowledge/ability, processes and experience on outcome measures (controlling for symptoms and IQ) will be conducted. G power estimation was used for a power calculation based on a power of 0.80, effect size of 0.313[60] and alpha of 0.05. This suggested for six predictors a total of 55 participants are required.

In terms of the mediation analysis, power estimation was calculated using the Monte Carlo method to estimate power for complex mediation models (see Thoemmes et al[95]). Using fixed parameters from Davies et al[60] and power at 0.8, this suggested a total sample size of 146 participants to detect mediation effects, outlined in the above model.

In terms of the longitudinal analysis, as many of the original sample (N=80) as possible will be followed up to maximise statistical power.

### Planned data analysis

Data will be double entered and checked for accuracy, and checked for outliers.

Missing data will be considered as 'Missing At Random' (MAR), which means the missing variables are related to additional observed variables within the data, but values of missing data itself. Missing data will be treated according to best practice.[96] Principled missing data methods will be used which combine available information from the observed data to estimate the population parameters and/or the missing data mechanism.[97] Full Information Maximum Likelihood (FIML) involves using all the observed data and creating values for missing data using maximum likelihood estimations. This works well provided that the model for the complete data is realistic.[98] FIML will be used within this study which is considered most appropriate for MAR data and for mediation analyses.[99] Quantitative data (including demographic information) will be reported using descriptive statistics, for example, means and SD.

Data will be analysed using SPSS and Mplus software. Group differences of all metacognitive measures and perceptual signal detection biases between young people with psychosis and healthy controls will be assessed, controlling for age and IQ. Linear regression analyses will be used to assess how metacognition (and perceptual signal detection biases) predicts anomalous experiences. Cross-sectional and longitudinal predictive analyses will be conducted, including regression analyses which will assess whether metacognitive components—knowledge/ability, experience and efficiency—are predicted by metacognitive processes. A mediation analysis will be used to check whether functional capacity, functional outcome and subjective recovery outcomes are predicted by metacognition and metacognitive processes, while controlling for anomalous experiences, symptoms and cognition. These analyses will include bootstrapped bias-corrected CI.

## DISCUSSION

Our assumption is that metacognitive variables predict, and maintain, anomalous events, poor objective and subjective recovery outcome in young people with and without psychosis. Particularly in young people with psychosis, metacognitive deficits may predict long-term functional outcome.

The study results will be an important addition to the literature and for clinicians for four main reasons: (i) this study tests a proposed model from previous literature which may help understand poor functional outcome in psychosis, (ii) from this novel intervention studies can be developed to tackle the potential metacognitive deficits in psychosis which predict this poor functional outcome, (iii) this is one of the first studies to assess metacognition as a longitudinal predictor of functional outcome in young people with psychosis sample, and finally, (iv) our studies use up-to-date paradigms within the field of metacognition to avoid biases.

### Limitations

A foreseeable limitation is that the FEP sample will comprise both a previous FEP sample and new participants who are currently engaged in early intervention services (EIS) in Sussex. Therefore, individuals will be at various stages of their recovery and support from the EIS which adds variation in terms of symptoms and recovery. With this in mind, symptoms will be controlled in the main analysis. However, this will enable exploration of factors which predict this variation. Due to length of follow-up period, another limitation may be the difficulty in re-recruiting these participants into the study. If recontacting is difficult, the model may not have full power. If so, the individual paths of the model in the longitudinal analyses will be explored.

## ETHICS AND DISSEMINATION

This study has been reviewed and approved by Camberwell St. Giles Research Ethics Committee (reference number: 17/LO/0055). The data will be stored securely in accordance with usual NHS procedures and data will be governed by the sponsor: University of Sussex. This research project will be reported within a PhD thesis, and will be written up for publication in scientific journals.

**Acknowledgements** The authors thank Dr Samuel Berens who assisted in the design and coding of the Matlab tasks. They thank the Patient and Public Involvement forum, Psychosis Theme Group (PTG), who reviewed and commented on the project, in order to provide a lived experience perspective. The Sussex Psychosis Research Interest Group and Sackler Centre for Consciousness Science have also reviewed the original protocol.

**Contributors** ACW designed the study, supervised by KG and DF.

**Funding** This work was supported by Sussex Partnership NHS Foundation Trust and Economic Social Research Council, through a PhD studentship awarded to the first author (Reference: ES/J500173/1).

**Competing interests** None declared.

**Patient consent** Not required.

**Ethics approval** Camberwell St. Giles Research Ethics Committee Committee (reference number: 17/LO/0055).

**Provenance and peer review** Not commissioned; externally peer reviewed.

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
