## [Reviewer comments · BMJ Open]

ARTICLE DETAILS

TITLE (PROVISIONAL)	Developing a dynamic model of anomalous experiences and function in young people with or without psychosis: a cross-sectional and longitudinal study protocol.
AUTHORS	Wright, Abigail; Fowler, David; Greenwood, Kathryn

VERSION 1 – REVIEW

REVIEWER	Jesus Perez University of Cambridge, UK
REVIEW RETURNED	26-Mar-2018

GENERAL COMMENTS	This is an excellent protocol for a cross-sectional study on meta-cognition and psychosis. The protocol is clearly described. The rationale is particularly useful to understand the background and potential use of this study to improve recovery and functioning in first-episode psychosis. I only have a few minor suggestions. First, it would be useful if the authors consistently use "anomalous" or "unusual" experiences throughout title, abstract and text. In methods, I found difficult to understand the composition of the sample. The authors refer to a previous study; is the sample going to be comprised of new cases and data from a previous study? Is the total sample 80 or 153? Do the authors envisage any limitation by applying this sample selection? Also, it might be informative to know the strategy to identify the 73 healthy volunteers. Other than this, I look forward to seeing this protocol published and, subsequently, the results from an interesting and novel study.
---

REVIEWER	Łukasz Gawęda II Department of Psychiatry, the Medical University of Warsaw, Poland
REVIEW RETURNED	07-May-2018

GENERAL COMMENTS	The authors presented a protocol of an interesting study attempting to understand the role of metacognition in anomalous experiences and functional outcomes in psychosis. I have no doubts that the object under investigation is important both for theory as well as for clinical practice. In general, methods proposed by the authors are well described. Longitudinal observation is an advantage of the study. My major concern refers to the theoretical background and rationale of the study. Concepts that the authors investigate and the linkage between them require careful theoretical consideration. In my opinion, the authors should improve the theoretical part of the study considerably starting
---

from more careful definitions and present the rationale of the study more clearly. This main part of the article seems chaotic. Please see below for my more detailed comments.

The review can enrich the rationale for the general relation between anomalous experiences and some metacognitive biases by discussing following papers: Nelson et al., 2014 (Two papers published in Schizophrenia Research).

Anomalous experiences is a very wide term referring to a rich number of various psychic phenomena. The authors should be more specific from the beginning of theoretical introduction about which anomalous experiences they investigate. It seems that the study is more focused on psychotic-like experiences (schizotypy, depersonalization). Furthermore, it is unclear and, in my opinion, unjustified, as the authors investigate anomalous experiences in psychosis, that only delusions, but not hallucinations are considered in the study. What is the rationale for that?. There are studies that addressed meta-cognition in hallucinations specifically showing no relation to delusions (e.g., Gaweda et al., 2013). What is more at the same time a line of studies have shown that meta-cognitive biases that may be not that modality-specific as the authors claim are linked to anomalous experiences in psychosis and are observed from the earliest phases of the illness (Gaweda et al., 2018, European Psychiatry). Furthermore, all these studies controlled for deficits in general performance.

It is not clear what is the difference between perceptual anomalies and perceptual biases (p.3). Please clear this point. Furthermore, the authors wrote: 'Perceptual biases are noted when an individual has a bias towards stating that a stimulus was present when it was in-fact absent.'. It is misleading, as there are a different kind of perceptual biases and it seems like the authors refer to false perception specifically or signal detection biases as some authors called (Algamani et al., 2017, Cognitive Neuropsychiatry).

Authors wrote 'It may be suggested that top-down biases predispose an individual to anomalous experiences.' To enrich a theoretical background, I strongly recommend to refer to studies showing the impact of top-down processes on (false) perception (see: e.g., Algamani et al., 2017; Fletcher and Frith, 2009, Nature Reviews Neuroscience; Cook et al., 2012, Frontiers in Human Neuroscience).

The part about modality-specificity of anomalous experience and metacognition is unclear and need further explanation. Majority of the core self-disturbances in psychosis are not modality specific, e.g., disturbances of the first-person perspective, hyper-reflexivity, lack of ego-boundaries, derealization, etc.).

The relationship between meta-cognition and functional outcomes is clearly explained. Please consider however anomalous experiences in the analyses. Anomalous experiences should be controlled along with symptoms.

Furthermore, as the authors target relationship between metacognition and both anomalous experiences and functional outcomes, I suggest to include also hypothesis 5 stating your expectations about the relationship between metacognitive variables and anomalous experiences at follow up.

Some minor comments:

'Metacognition is defined as "thinking about thinking" (Semerari et al., 2003) and the way one thinks about one's experience.' (p. 3). First, I suggest citing Flavell's work, as he introduced the term of metacognition. The second part of the sentence doesn't refer to metacognition (for instance 'I am worthless' is the way I am thinking about my experience of failing the exam, but it is object-level

	thinking according to Nelson and Narens). There are missing citations in some sentences (e.g., 'Such experiences are suggested to be a precursor to full psychotic symptoms, e.g., hearing voices, and increasingly distressing experiences.' p. 2) Literature has demonstrated that hallucinatory experiences and delusional beliefs/ideation have been associated with metacognition (overconfidence, specifically) in both clinical and non-clinical groups (Eisenacher et al., 2015; Moritz et al., 2005; Warman, 2008).' As I mentioned before, there are other studies referring directly to the discrimination between imagination and reality and metacognition (confidence) in psychosis (Gaweda et al., 2012; 2013; 2014; 2018). Some terms or definitions are uncommon: e.g., 'Anomalous delusional beliefs are unusual' I am not sure to what anomalous is referred here. Are there 'normal' delusional beliefs, or perhaps authors refer to the bizarreness of some delusional beliefs? 'Anomalous delusional beliefs, such as paranoia, are commonly experienced by those with psychosis, those considered at high-risk, and those within the general population (Hodgekins et al., 2012; Lower et al., 2015).' Please be more specific. It is misleading to say that all these groups commonly experience paranoia. We see extended phenotype (continuum), but at very different levels of expression. '(...) that derealisation (anomalous self-experience) may be associated with metacognitive deficits. This study will empirically test this. Anomalous self-experiences have not been considered to be modality-specific. Hence this has not been reflected in the construction of the scales, and will therefore not be assessed here.' I am not sure what will and what will not be tested. Participants: Exclusions criteria should be described in a more detailed manner (substances, controls: how mental health will be screened, family history of psychosis, etc?). Methods: The authors may also consider adding the Anomalous subscale Experiences of the SSI. Experimental tasks are well designed and described. Data analysis strategy is clear. Finally, potential shortcomings of the study should be recognized and discussed.
--	--

VERSION 1 – AUTHOR RESPONSE

Replies to reviewer 1 comments:

1. First, it would be useful if the authors consistently use "anomalous" or "unusual" experiences throughout title, abstract and text.

Thank you for pointing this out. We have changed this term to anomalous experiences throughout the article, abstract and title.

2. In methods, I found difficult to understand the composition of the sample. The authors refer to a previous study; is the sample going to be comprised of new cases and data from a previous study? Is the total sample 80 or 153?

We have clarified this point: “Seventy-three young people with psychosis will be recruited. This sample will be made up of a convenience sample from first episode services and the remaining individuals will be those re-recruited from a previous first episode psychosis (FEP) sample (previous N=80), to take part in the main cross-sectional study and longitudinal follow-up aspect.”

3. Do the authors envisage any limitation by applying this sample selection?

We acknowledge the limitations with this sample selection within a new ‘limitations’ section in the discussion: “A foreseeable limitation is that the First Episode Psychosis sample will be comprised of both a previous FEP sample and new participants who are currently engaged in EIS in Sussex. Therefore, individuals will be at various stages of their recovery and support from the Early Intervention Services which adds variation in terms of symptoms and recovery. With this in mind, symptoms will be controlled in the main analysis. However, this will enable exploration of factors which predict this variation.”

4. Also, it might be informative to know the strategy to identify the 73 healthy volunteers.

We have clarified this, alongside the screening of mental health difficulties. “Participants will be recruited through advertisement within the local community, e.g. in libraries, cafes and on social media. Participants with current mental health problems or history of psychosis will be excluded following screening questions: i) Are you currently experiencing any mental health difficulties? ii) Are you on any psychotropic medication/substances? iii) Have you been in contact with psychological or psychiatric services for psychological problems? iv) Has anyone in your immediate family experienced an episode of psychosis? E.g. parents, siblings. If healthy control answered yes to any of the questions, these participants were deemed ineligible to take part in the study”.

Reply to reviewer 2 comments:

1. The review can enrich the rationale for the general relation between anomalous experiences and some metacognitive biases by discussing following papers: Nelson et al., 2014 (Two papers published in Schizophrenia Research).

Thank you for this comment. This has been added to include these references and to clarify several points below. “Many theories have been proposed to understand anomalous experience, including source-monitoring deficit (Nelson et al. 2014a); difficulties in the internal monitoring and comparator system (Frith, 1987), or aberrant salience hypothesis (Nelson et al., 2014b); difficulty in failing to suppress attention to irrelevant or familiar information (Hemsley, 1993). Signal detection theory (SDT) has been the foundation to this research and many studies using SDT have demonstrated that anomalous perceptual experiences are associated with perceptual signal detection biases (Bentall & Slade, 1985; Kok, Kouider, Lange, & Supe, 2015); bias towards stating that a stimulus was present when it was in-fact absent. Such signal detection biases have been consistently demonstrated within the psychosis literature (Mussgay & Hertwig, 1990; Teufel et al., 2015; Teufel, Kingdon, Ingram, Wolpert, & Fletcher, 2010) and demonstrate that top-down processes on (false) perception can lead to experiences of hallucinations or delusions (Algamani et al., 2017; Cook et al., 2012; Fletcher and Frith, 2009; Powers et al., 2017).”

2. Anomalous experiences is a very wide term referring to a rich number of various psychic phenomena. The authors should be more specific from the beginning of theoretical introduction about which anomalous experiences they investigate. It seems that the study is more focused on psychotic-like experiences (schizotypy, depersonalization).

We have changed the wording to accommodate this comment: “Anomalous experiences refer to a

rich number of various psychic phenomena. These experiences can include the sense that you are not “real” (anomalous self-experiences; distortions in experience of self and being), hearing sounds which cannot be accounted for by the environment (anomalous perceptual experiences: distortions of sensory events in various domains; auditory, visual; touch; taste) and experiencing unusual thoughts or beliefs (anomalous delusional beliefs). These experiences may be commonly experienced by individuals within the general population (Bell, Halligan, & Ellis, 2006; Kelleher et al., 2012) but the frequency and intensity of these anomalous experiences is increased in those with psychosis, or those with emerging severe mental health difficulties (Brett, Johns, Peters, & McGuire, 2009; Reininghaus et al., 2016) and may be marker for psychotic vulnerability or predict psychotic symptoms at a later stage (Nelson, Sass and Skodlar, 2009; Nelson, Parnas & Sass, 2014; Yung et al., 2003; Yung, Phillips, Yuen, & McGorry, 2004)”

3. Furthermore, it is unclear and, in my opinion, unjustified, as the authors investigate anomalous experiences in psychosis, that only delusions, but not hallucinations are considered in the study. What is the rationale for that?

Thank you for pointing this out. We have included an assessment of low-level hallucinations by using a measure of anomalous perceptual experiences on MUSEQ which has previously been validated against CAPS, O-LIFE (Mitchell et al., 2017). We have used this measure as we were able to capture anomalous perceptual experiences in two modalities – visual and auditory.

We have clarified that we are using a measure of hallucinatory experiences by including the phrase “However, anomalous perceptual-experiences (e.g. hallucinations) in psychosis, and in the general population, can vary in modalities (Mitchell et al., 2017)”.

From your suggestion below, we have also now included the SSI anomalous experience scale to confirm the MUSEQ scores, and their association with perception and metacognition, as this is a well-known measure of anomalous experiences.

4. There are studies that addressed meta-cognition in hallucinations specifically showing no relation to delusions (e.g., Gaweda et al., 2013). What is more at the same time a line of studies have shown that meta-cognitive biases that may be not that modality-specific as the authors claim are linked to anomalous experiences in psychosis and are observed from the earliest phases of the illness (Gaweda et al., 2018, European Psychiatry).

Thank you for your detailed comments about this. We acknowledge that previous work has assessed verbal vs. non-verbal performed/imagined actions in FEP and UHR with no difference of misattribution between verbal or non-verbal action. This suggests that metacognition may be deficit across modalities for those with FEP and at-risk. However, we acknowledge that anomalous perceptual experiences and perceptual/metacognitive ability can be different for individual people, in terms of the modalities of experiences (Mitchell et al., 2017) alongside modality-specific metacognitive ability (measured using meta-d') (Fleming et al., 2014; Morales, Lau & Fleming, 2017). In addition, we acknowledge that there is a subgroup of people in psychosis who display intact cognitions (Greenwood et al., 2005), but still experience psychotic symptoms. In addition, auditory anomalous experiences are most common in psychosis (Shergill, Murray, & McGuire, 1998; Waters, Allen, et al., 2012) and may therefore demonstrate differential association with auditory perception or metacognition. Therefore, we are particularly interested in testing whether there is a direct relationship between specific perceptual/metacognitive abilities and perceptual anomalous experiences in the same modality: visual and auditory. This will be tested in both FEP and healthy control to capture the variation in anomalous experiences and metacognition.

We have amended the section: “A recent controlled study demonstrated that individuals with FEP and those at-risk were more likely to misattribute an imagined action for a performed action, compared to healthy controls (Gaweda et al., 2018), but as there was no difference in misattribution of verbal or non-verbal actions this suggests the deficit in metacognition may be across several modalities. However, metacognitive ability (measured using meta-d') is known to be modality-specific (Fleming et

al., 2014; Morales, Lau & Fleming, 2017) and anomalous perceptual-experiences (e.g. hallucinations) can vary in modalities (Mitchell et al., 2017). It has also been acknowledged that auditory anomalous experiences are most common in psychosis (Shergill, Murray, & McGuire, 1998; Waters, Allen, et al., 2012), all of which may suggest a modality-specific association with auditory or visual anomalous experiences and perception/metacognition. This present study will assess the modality-specific association between perceptual bias (signal detection bias) and metacognitions with anomalous perceptual experiences in visual and auditory modalities, whilst controlling for objective performance (see figure 1).”

5. Furthermore, all these studies controlled for deficits in general performance.

We acknowledge that some studies have controlled for performance and we have noted this hence the inconsistency in the literature when performance is controlled (Gaweda et al., 2013). We have now included the phrase: “Some studies did not control for objective performance”.

6. It is not clear what is the difference between perceptual anomalies and perceptual biases (p.3).

Please clear this point.

Please see point 1.

7. Furthermore, the authors wrote: ‘Perceptual biases are noted when an individual has a bias towards stating that a stimulus was present when it was in-fact absent.’ It is misleading, as there are a different kind of perceptual biases and it seems like the authors refer to false perception specifically or signal detection biases as some authors called (Algamani et al., 2017, Cognitive Neuropsychiatry).

Thank you for the recommendation. We have changed this term to “signal detection biases” and included more information on this in response to point 1.

8. Authors wrote ‘It may be suggested that top-down biases predispose an individual to anomalous experiences.’ To enrich a theoretical background, I strongly recommend to refer to studies showing the impact of top-down processes on (false) perception (see: e.g., Algamani et al., 2017; Fletcher and Frith, 2009, Nature Reviews Neuroscience; Cook et al., 2012, Frontiers in Human Neuroscience).

Please refer to point 1 response.

9. The part about modality-specificity of anomalous experience and metacognition is unclear and need further explanation. Majority of the core self-disturbances in psychosis are not modality specific, e.g., disturbances of the first-person perspective, hyper-reflexivity, lack of ego-boundaries, derealization, etc.).

From this, we have altered sentence: “Anomalous self-experiences have not been considered to be modality-specific, hence this has not been reflected in the construction of the scales, and these self-experiences are hypothesised to be related to both visual and auditory perceptual biases and metacognitive ability”.

10. The relationship between meta-cognition and functional outcomes is clearly explained. Please consider however anomalous experiences in the analyses. Anomalous experiences should be controlled along with symptoms.

Added into hypothesis: ‘Hypothesis 3: Metacognitive variables (metacognitive knowledge, metacognitive processes and metacognitive experience) will significantly predict outcome measures (functional capacity, functional outcome and subjective recovery outcome) in young people with and without psychosis, even after controlling for anomalous experiences, symptoms and IQ’.

The analysis section has also been changed accordingly: ‘A mediation analysis will be used to whether functional capacity, functional outcome and subjective recovery outcome are predicted by metacognition and metacognitive processes, whilst controlling for anomalous experiences, symptoms and cognition’.

11. Furthermore, as the authors target relationship between metacognition and both anomalous experiences and functional outcomes, I suggest to include also hypothesis 5 stating your expectations about the relationship between metacognitive variables and anomalous experiences at follow up. Thank you for this comment on the hypotheses/analysis. Unfortunately, we are unable to include this hypothesis as we do not have anomalous experience measures at time-point one. Hence we are controlling for symptoms as a proxy of this using PANSS. We hope this is sufficient.

Minor comments:

12. 'Metacognition is defined as "thinking about thinking" (Semerari et al., 2003) and the way one thinks about one's experience.' (p. 3). First, I suggest citing Flavel's work, as he introduced the term of metacognition.

We have added this.

13. The second part of the sentence doesn't refer to metacognition (for instance 'I am worthless' is the way I am thinking about my experience of failing the exam, but it is object-level thinking according to Nelson and Narens).

We have removed the end part of the sentence for clarity.

14. There are missing citations in some sentences (e.g., 'Such experiences are suggested to be a precursor to full psychotic symptoms, e.g., hearing voices, and increasingly distressing experiences.' p. 2)

We have altered this paragraph: "...the frequency and intensity of these anomalous experiences is increased in those with psychosis or those with emerging severe mental health difficulties (Brett, Johns, Peters, & McGuire, 2009; Reininghaus et al., 2016) and may predict psychotic symptoms at a later stage (Yung et al., 2003; Yung, Phillips, Yuen, & McGorry, 2004)".

15. Literature has demonstrated that hallucinatory experiences and delusional beliefs/ideation have been associated with metacognition (overconfidence, specifically) in both clinical and non-clinical groups (Eisenacher et al., 2015; Moritz et al., 2005; Warman, 2008). As I mentioned before, there are other studies referring directly to the discrimination between imagination and reality and metacognition (confidence) in psychosis (Gaweda et al., 2012; 2013; 2014; 2018).

Thank you for these references. We have added these into the manuscript.

16. Some terms or definitions are uncommon: e.g., 'Anomalous delusional beliefs are unusual' I am not sure to what anomalous is referred here. Are there 'normal' delusional beliefs, or perhaps authors refer to the bizarreness of some delusional beliefs?

I have used this term as it fits well with the anomalous experience/self-experience aspect to demonstrate that we are looking at perceptual experiences, self-experiences and unusual beliefs and their association with metacognition. I have included the example, e.g. paranoia, to highlight that I am focusing on common delusional beliefs.

17. Anomalous delusional beliefs, such as paranoia, are commonly experienced by those with psychosis, those considered at high-risk, and those within the general population (Hodgekins et al., 2012; Lower et al., 2015). Please be more specific. It is misleading to say that all these groups commonly experience paranoia. We see extended phenotype (continuum), but at very different levels of expression.

We have since deleted this phrase as we have shortened the introduction.

18. '(...) that derealisation (anomalous self-experience) may be associated with metacognitive deficits. This study will empirically test this. Anomalous self-experiences have not been considered to be modality-specific. Hence this has not been reflected in the construction of the scales, and will

therefore not be assessed here.' I am not sure what will and what will not be tested.
Please see response to point number 9.

19. Exclusions criteria should be described in a more detailed manner (substances, controls: how mental health will be screened, family history of psychosis, etc?).
This section has been altered: "A lack of mental health condition was confirmed using four screening questions: i) Are you currently experiencing any mental health difficulties? ii) Are you on any psychotropic medication/substances? iii) Have you been in contact with psychological or psychiatric services for psychological problems? iv) Has anyone in your immediate family experienced an episode of psychosis? E.g. parents, siblings. If healthy control answered yes to any of the questions, these participants were deemed ineligible to take part in the study."
In the clinical participant aspect: "Participants with organic causes for psychosis or those with a diagnosis of substance use disorder will be excluded."

20. The authors may also consider adding the Anomalous subscale Experiences of the SSI.
We thank the reviewer for this comment as it is a useful point. The anomalous experience subscale as a measure of anomalous perceptual experiences, to confirm MUSEQ data. Please see point 3.

21. Finally, potential shortcomings of the study should be recognized and discussed.
A limitations section has been added p.16: "One foreseeable limitation is that the First Episode Psychosis sample will be comprised of both a previous FEP sample and new participants who are currently engaged in EIS in Sussex. Therefore, individuals will be at various stages of their recovery and support from the Early Intervention Services which adds variation in terms of symptoms and recovery. With this in mind, symptoms will be controlled in the main analysis. However, this will enable exploration of factors which predict this variation. Due to length of follow-up period, another limitation may be the difficulty in re-recruiting these participants into the study. If re-contacting is difficult, the model may not have full power. If so, the individual paths of the model in the longitudinal analyses will be explored."

VERSION 2 – REVIEW

REVIEWER	Jesus Perez University of Cambridge
REVIEW RETURNED	09-Jul-2018

GENERAL COMMENTS	No further comments.
----------------------

REVIEWER	Łukasz Gawęda II Department of Psychiatry, the Medical University of Warsaw, Poland
REVIEW RETURNED	28-Jul-2018

GENERAL COMMENTS	The authors did a good job in revising the paper. I think it is ready for publication.
--